# Developmental Regression Followed by Epilepsy and Aggression: A New Syndrome in Autism Spectrum Disorder?

**DOI:** 10.3390/jpm13071049

**Published:** 2023-06-26

**Authors:** John Gaitanis, Duyu Nie, Tao Hou, Richard Frye

**Affiliations:** 1Hasbro Children’s Hospital, The Warren Alpert Medical School of Brown University, Providence, RI 02903, USA; dnie@lifespan.org; 2Department of Nutrition, Harvard T.H. Chan School of Public Health, Boston, MA 02115, USA; taohou@gmail.com; 3Autism Discovery and Treatment Foundation, Phoenix, AZ 85050, USA; drfrye@rossignolmedicalcenter.com; 4Rossignol Medical Center, Phoenix, AZ 85050, USA

**Keywords:** autism spectrum disorder with regression, epilepsy, aggression, aggressive behaviors, self-injurious behaviors

## Abstract

Autism spectrum disorder (ASD) with regression (ASD-R) involves the loss of previously attained developmental milestones, typically during the first or second year of life. As children age, it is not uncommon for them to develop comorbid conditions such as aggressive behaviors or epilepsy, which can inhibit habilitation in language and social function. In this paper, we hypothesize that aggressive behaviors and epilepsy more commonly develop in patients with ASD-R than in those without a history of regression (ASD-NR). We conducted a retrospective review of non-syndromic patients with ASD over 12 years of age and compared the rates of epilepsy and aggression between ASD-R and ASD-NR patients. Patients with ASD-R, as compared to ASD-NR patients, demonstrated non-significantly higher rates of epilepsy (51.8% vs. 38.1%, *p* = 0.1335) and aggressive behaviors (73.2% vs. 57.1%, *p* = 0.0673) when evaluated separately. The rates for combined epilepsy and aggression, however, were statistically significant when comparing ASD-R versus ASD patients (44.5% vs. 23.8%, *p* = 0.0163). These results suggest that epilepsy with aggression is more common in ASD-R as compared to ASD-NR patients. When considering the impact of epilepsy and aggression on quality of life, these co-morbidities effectively cause a second regression in patients who experienced an earlier regression as toddlers. A larger, prospective trial is recommended to confirm these associations and further define the timeline in which these characteristics develop from early childhood to adolescence.

## 1. Introduction

Autism spectrum disorders (ASD) are accompanied by multiple comorbid conditions [1]. Among the most common are anxiety disorders, sleep disturbances, gastrointestinal disorders, and feeding intolerance. Aggressive behaviors occur in more than half of ASD patients [2] and can result in social isolation [3] and heightened parental stress [4]. Epilepsy is also commonly associated with ASD and adversely impacts cognition and increases mortality risk [5,6]. The accurate prediction of epilepsy and aggressive behaviors holds the promise of intervention prior to their onset. This study hypothesizes that epilepsy and aggression are more common in patients who had a developmental regression in early childhood and that these co-morbidities usually co-occur.

### 1.1. ASD with Regression

ASD with regression (ASD-R) involves the loss of previously attained developmental milestones [7], typically during the first or second year of life [8,9,10,11]. The average age of regression is around 20 months [12]. Loss of language is the most frequently reported symptom [13] and typically affects children with pre-existing language delay. Kurita [9] found that 94% of children used only single words and had a limited vocabulary at the time of their speech regression. A decline in social skills can also occur and involves the loss of eye contact and imitative games [13].

The causes of regression in ASD are not understood but are likely multi-factorial. Epilepsy [14] and sleep-activated epileptic discharges (i.e., Landau–Kleffner syndrome) [11] do not appear to have a causal role in most cases. By contrast, a family history of autoimmune thyroid disease may be associated with ASD-R [15]. Children with ASD-R have higher levels of ASD symptomology [16,17] and higher rates of moderate to severe intellectual disability compared to ASD patients without regression (ASD-NR) [17,18,19].

### 1.2. Epilepsy Associated with ASD

Epilepsy and ASD are both heterogeneous disorders caused by multiple different etiologies, some of which are common to both conditions. As many as one-third of children with epilepsy are at risk of having an ASD—the risk is higher in those with seizure onset at a younger age [20,21]. Similarly, in children with ASD, as many as one-third have epilepsy [22,23,24], with the prevalence increasing with age. Indeed, Viscidi [25] demonstrated that the rates of epilepsy in children with ASD aged between 2–17 years was 12.5%, compared to 26% for children with ASD aged 13 years and older. Therefore, any evaluation of ASD and epilepsy needs to include older age groups to avoid underestimating the prevalence.

Cognitive dysfunction, motor deficits, and severe receptive language impairment are more commonly seen in patients with epilepsy and ASD [26]. When these conditions co-exist, the question arises whether epileptiform discharges directly contribute to the developmental impairments. The term epileptic encephalopathy is used when “the epileptiform abnormalities themselves are believed to contribute to the progressive disturbance in cerebral function” [27]. Landau–Kleffner syndrome (LKS) and epilepsy with continuous spike waves during slow-wave sleep (CSWS) are the two forms of epileptic encephalopathy most frequently implicated in ASD. LKS occurs after 3 years of age, in association with an epileptiform EEG that is predominantly over the temporal regions. In approximately 25% of children, LKS occurs without clinical seizures [28]. In LKS, epileptiform discharges occurring in areas serving language function are implicated in causing aphasia [29]. Likewise, in tuberous sclerosis complex, tubers in the temporal lobes and temporal lobe epileptiform discharges predispose patients to ASD [30].

Although both LKS and ASD-R feature language regression as a primary symptom, the age of regression differs. Children with ASD-R experience regression at a younger age, generally before 2 years. By comparison, patients with LKS have a regression in language after 3 years of age [31,32,33]. Children with ASD and late-onset regression after age 3 are classified in the subgroup of disintegrative disorder. These patients have a higher incidence of epilepsy when compared to other patients with ASD (70% vs. 30%) [34,35]. The exact relationship between ASD-R, LKS, and childhood disintegrative disorder is not well-delineated. They share regression as a core feature, but the specific rate of epilepsy in ASD-R has not been well-studied, despite clear similarities to LKS and childhood disintegrative disorder.

### 1.3. Autism and Behavioral Aggression

The overall prevalence of aggression in ASD is 53.7% [2], with as many as 68% of individuals exhibiting physical aggression toward caregivers [36]. Aggressive behaviors can continue into adulthood in 15 to 18% of patients with ASD and intellectual disability [37]. Contributors to aggression include self-injurious behavior, ritualistic behavior, and resistance to change [36]. Self-injurious behaviors are also common in ASD, with prevalence rates ranging from 30% to 53% [38,39,40,41]. Self-injurious behaviors are significant precursors of later aggression among children with intellectual disability [42]. When evaluating co-morbid conditions that relate to aggression, sleep disturbance has a strong association [43,44], whereas gastrointestinal disturbance does not appear to be related [45].

Families caring for children with ASD and aggression report unique challenges. They describe social isolation, concerns for safety, and a lack of respite care and other professional supports. As children age, aggressive behaviors may limit their options for residential housing. Parents report an “unbearable” level of exhaustion, with one describing her situation as being in “jail for life” [3]. Finding improved treatment modalities therefore benefits both the patient and their family.

Despite the high prevalence of aggression in ASD, there are no studies that have compared the prevalence of aggression in patients with ASD-R versus ASD-NR. Conversely, ASD-R patients have been found to have a heightened risk of epilepsy [25]. This study hypothesizes that patients with ASD-R are more likely to develop epilepsy and aggression than patients with ASD-NR. Since regression typically develops at a young age (less than 2 years), identifying an elevated risk of co-existing epilepsy or aggression will help clinicians to anticipate these concerns before they develop and potentially take action to mitigate their development. Identifying a strong association between ASD-R, epilepsy, and aggression may also clarify if they share etiologic underpinnings, and perhaps respond to similar treatment pathways. Since epilepsy has a higher prevalence in ASD patients over 13 years of age [25], and aggression may occur in adulthood [37], this study only examined patients 12 years of age and over. Younger patients were excluded, since their inclusion would have risked underestimating the true prevalence of these co-morbidities. Patients with known genetic syndromes were also excluded (i.e., Dravet syndrome, tuberous sclerosis complex), since they have distinct clinical features which may have skewed the results. This evaluation therefore focused on non-syndromic patients with ASD of 12 years of age and over.

## 2. Materials and Methods

This study was conducted in accordance with the Declaration of Helsinki and the Institutional Review Board of Tufts Medical Center (the senior author’s primary institution at the time of the study). IRB approval was granted for a retrospective chart review of patients with ASD (IRB # 13200). Patients were identified from the records of the pediatric neurology division at Tufts Medical Center. A chart review was conducted by searching for all patients aged 12 years or over with a diagnostic code for ASD. The absence or presence of regression, epilepsy, and aggression was recorded from the medical records, as were the ages that these conditions developed. Data on sleep disruption were also recorded.

Patients with ASD-R were defined as having a loss of any previously acquired communication or social skill [25]. To be counted as having ASD-R, patients required a history of either language regression (i.e., loss of verbalizations or words), social regression (i.e., loss of eye contact or joint attention), or both. Every patient was asked about a history of regression and was recorded as having ASD-R if either language or social regression was reported. To qualify as having regression, a patient had to demonstrate a previously higher level of skills before their decline. The nadir of developmental progress had to occur after clearly establishing greater functioning. The medical history required clear descriptions of these declines.

One example of such a history was taken from a boy who experienced a regression at 16 months of age. Quoting his medical records, “he was making appropriate eye contact and was babbling. He was also saying ‘mamma.’ He was very social and interactive. He was affectionate towards his parents. The regression occurred rapidly, occurring ‘overnight.’ At approximately 16 months, he lost eye contact and stopped speaking. He was no longer babbling or jabbering. Most of his vocalizations consisted of crying and screaming. He began to mouth objects. He did not want to be touched and wanted reduced stimulation.”

A separate patient reported a similar loss of function at approximately 12 months of age. In his chart, it was reported that “prior to 12 months, he made excellent eye contact and was very social and interactive. He waved bye and said “mama,” “dada,” “hi,” and “bye.” After 12 months, he suddenly became more silent and no longer babbled. He became completely quiet and no longer vocalized at all.”

Since this was a retrospective analysis, video evidence of regression was not uniformly available and was therefore not required when defining ASD-R. A history of developmental delays prior to regression did not exclude patients from having ASD-R. Rather, ASD-R was defined as a loss of function but did not require age-appropriate developmental progress prior to the decline in functioning.

Epilepsy was characterized as recurrent, unprovoked seizures (ILAE 1993). Seizures following a clear trigger (i.e., febrile seizures) were not counted as epilepsy. Clinical evidence of recurrent clinical seizures was required for a diagnosis of epilepsy, but EEG or MRI abnormalities were not. All seizure types (generalized and focal) were included. Epilepsy syndromes (i.e., infantile spasms, childhood absence epilepsy) were also included as epilepsy and were recorded when appearing in the records.

Aggression was defined as acts of physical harm [2]. Given the association between self-injurious behaviors (SIB) and aggression [46], acts of physical harm to both oneself and others were included. The age that such behaviors developed was recorded. Behavioral challenges not including physical harm to oneself or others were not counted as aggression (i.e., defiance, freezing, obsessive compulsive behaviors, eloping). One example of aggressive behavior taken from the medical records describes a teenage boy who “will bang his head on the wall or hit his head with knuckles. If others try to stop him from hitting himself, he will become aggressive towards them.”

Sleep disturbances included insomnia (impairments of sleep initiation, duration, consolidation, or quality), sleep-related breathing disorders, and sleep-related movement disorders [47]. A history of daytime fatigue, consultation with a sleep specialist, and a formal sleep study were not required to make a diagnosis of a sleep disturbance.

Statistical analyses were conducted using SAS version 9.4 (SAS Institute Inc., Cary, NC, USA). Categorical variables are presented as percentages, and continuous variables were expressed as averages with the interquartile range (IQR, between the 25th percentile and 75th percentile). We compared the frequency of the clinical variables between the ASD-NR and ASD-R cohorts with the chi-square test for categorial variables and the one-tailed t-test for continuous variables. A *p*-value < 0.05 was considered statistically significant.

## 3. Results

A total of 134 patients with ASD over 12 years of age were referred for neurological evaluation (Table 1). A total of 16 were excluded due to the presence of a confirmed genetic diagnosis, leaving a total of 119 included in the study. Among these, 56 had a history of ASD-R and 63 had ASD-NR. Among those patients, 17 were female and 102 were male.

The prevalence of epilepsy was higher in ASD-R (51.8%) as compared to ASD-NR (38.1%), though not reaching a statistic significance (*p* = 0.1335). There was no significant difference in the mean age of epilepsy onset (ASD-R: 7.83 (IQR 3.00–12.50) years; ASD-NR: 5.26 (IQR 1.20–10.00) years). The mean age of ASD diagnosis was significantly younger in ASD-R (2.46 (IQR 1.94–2.75) years) as compared to ASD-NR (3.98 (IQR 2.00–5.00) years; *p* = 0.001).

The prevalence of aggressive or self-injurious behaviors (SIB) seemed to be higher in ASD-R patients (73.2%) as compared to ASD-NR patients (57.1%), though not reaching a statistic significance (*p* = 0.0673). There was no significant difference in the mean age of onset of aggression or SIB (ASD-R: 9.40 (IQR 7.00–12.00) years; ASD-NR: 10.07 (IQR 7.00–14.00) years). The prevalence of sleep disturbance was similar in ASD-R versus ASD-NR cohorts (52.4% versus 60.7%, *p* = 0.3604).

Of the 119 patients studied, 33.6% (40/119) had both epilepsy and aggression with the prevalence significantly higher in ASD-R (44.6%) as compared to ASD-NR (23.8%; *p* = 0.0163). This indicates a significantly higher prevalence of comorbid epilepsy and aggression in children with ASD-R in comparison to ASD-NR. The relative risk (RR) was 1.59 (95% confidence interval 1.10–2.29).

## 4. Discussion

This study explores the relationship between ASD-R and the co-morbidities of epilepsy, sleep disturbance, and aggression. Since epilepsy and aggression may not develop until pre-adolescence to adolescence, only patients 12 years of age and older were evaluated. In this retrospective review, epilepsy and aggression were found to be more common in patients with ASD-R than in patients with ASD-NR, although those differences were not statistically significant for each comorbidity alone. However, there was indeed a significantly higher rate of patients with both epilepsy and aggression in the ASD-R group than in the ASD-NR group. The majority first developed regression and then epilepsy and, later, aggressive behaviors.

When considering the impact of epilepsy and aggression on quality of life, these co-morbidities effectively cause a second regression in patients who experienced an earlier regression as toddlers. Epilepsy carries medical risks and can have a negative impact on cognition [5,6]. Similarly, aggression has deleterious impacts on daily routines and the well-being of family members [3]. Both conditions have a dramatic impact on the functioning of children with ASD who are already struggling to advance. That the two conditions can occur in the same child, with 44.6% of patients with ASD-R developing both epilepsy and aggression, speaks to the seriousness of this symptom complex.

These findings suggest that epilepsy and aggression are more common in children with ASD-R than in those with ASD-NR, although a larger, prospective trial is needed to confirm such an association. They also demonstrate that these co-morbidities develop years after the initial developmental regression and diagnosis of ASD (a mean of 2.46 years old at onset in patients with ASD-R). The characteristic sequence is a developmental regression as a toddler, followed by epilepsy at elementary school age, and finally, aggressive behaviors in pre-adolescence. The association between these conditions suggests a shared etiology whose timing may be based on ontogeny. It emphasizes the importance of following patients throughout their life cycle to fully understand the risk of developing such co-morbidities.

There are multiple mechanisms through which autism, epilepsy, and aggression may co-develop. Genetic etiologies are necessary initial considerations. Epileptic encephalopathies are equally important to explore, but since the regression can occur either before, during, or after the development of epilepsy, epileptic encephalopathy will not account for all cases of regression [48]. Other conditions worth evaluating include inborn errors of metabolism, systemic inflammatory disorders, toxic exposures, and endocrinopathies. A detailed medical assessment, considering a broad range of pathologies, is therefore essential.

When initiating a genetics assessment, numerous potential etiologies must be explored. The history and physical examination should guide the work-up before initiating laboratory investigations. The multiple genetic conditions causing autism and epilepsy exert their effects through varied pathophysiologic mechanisms. Rett syndrome, for example, disrupts synaptic development [49]. Similarly, tuberous sclerosis causes synaptic impairment [50] but also results in the disordered regulation of cellular growth [51]. Developmental regression, autism, and epilepsy can also be seen in disorders of ion channels (i.e., Dravet syndrome), impairments of receptor expression (i.e., GRIN1), transcription factors (i.e., *MEF2C*), axonal guidance (i.e., *NTNG1*), and ubiquination (i.e., *RHOBTB2*) [48]. Hence, the genetic mechanisms contributing to a phenotype of regression, autism, and epilepsy are diverse. Specific genetic syndromes such as these were not included in our analysis, since they each carry unique rates of autism and epilepsy that are not necessarily representative of the broader population. Still, this diversity of pathophysiology informs us of the potential mechanisms in non-syndromic patients and is illustrative of potential treatment avenues that might be relevant for all patients.

Metabolic conditions causing autism and epilepsy are equally diverse. Mitochondrial disorders can result in autistic regression and epilepsy [52]. Likewise, homocystinuria, a disorder of amino acid metabolism, when untreated causes symptoms of ASD [53] and seizures [54]. Cerebral folate deficiency is characterized by abnormally low cerebral spinal fluid folate levels despite normal serum values and results in sleep disturbances, epilepsy, developmental regression, autism, ataxia, and extrapyramidal symptoms [55]. The primary transporter for folate across the blood–brain barrier is the folate receptor *α*, which is blocked by cerebral folate receptor autoantibodies in as many as 60% of children with ASD [56]. This was the basis for a prospective open-label trial of d,l-leucovorin, a reduced folate that can cross the blood–brain barrier [57]. In that trial, treatment with leucovorin improved receptive and expressive language in about two-thirds of patients. Such findings highlight the potential for leveraging knowledge of etiologies towards personalized treatment avenues.

Autism susceptibility is estimated to be 40–80% genetic [58], with environmental influences presumed to account for the remainder. Genetic susceptibility interacts with pre-natal exposures through genes to environment interactions [59]. Numerous early life environmental factors are implicated, including advanced parental age, maternal infections during pregnancy, and prenatal exposure to anti-seizure medications [60,61,62]. Some environmental contributors have been suggested for both ASD and epilepsy. Maternal stress is one potential contributor to both conditions [59,63]. Anxiety alters levels of stress hormones and can induce inflammation. Maternal immune activation can also be triggered by viral or bacterial infections during pregnancy, causing a cytokine storm that interferes with proper development of the fetal brain [64]. This mechanism has been implicated in the development of both autism [65] and epilepsy [64].

This interaction between the nervous and immune systems is also implicated in SIB, which shares similar features to neuropathic pain [66], in which hyperalgesia is mediated through inflammatory, immune, and nociceptive systems. The endogenous opioid system is disrupted in ASD [67], possibly contributing to SIB. Endogenous opioids play a role in social development [67], restlessness, and hyperactivity [68]. Endogenous opioids also promote the expression of cytokines, including interleukin-6 (IL-6) and IL-1 [69,70]. Similar cytokine expression is observed in drug-resistant epilepsy [71]. This suggests a potential role for immunomodulatory therapies that may improve both SIB and epilepsy in ASD patients.

When epilepsy and aggression co-exist, the direct role of seizures, and their treatment, must also be explored. Aggressive behavior is generally unrelated to the seizures themselves [72]. The incidence of aggressive conduct during a seizure has an estimated incidence of only 1 in 1000 [73]. Violent-appearing movements during ictal phenomena are non-purposeful and not directed in a conscious fashion, distinguishing them from volitional behaviors [74]. Rather, aggression more frequently occurs as a feature of post-ictal psychosis [75]. Continuous spike waves in slow sleep can also contribute to behavioral alterations [76]. When aggressive behaviors occur in persons with epilepsy, seizures frequently localize to the frontal lobe [77]. Beyond the seizures themselves, the role of medications in inducing aggressive behaviors should be considered. A questionnaire survey of children and teenagers revealed that 50% reported feeling “cross/irritable” and 30% were “angry” as a result of their anti-seizure medication [78]. Benzodiazepines are associated with behavioral disturbance in as many as 15% of patients [79]. Lamotrigine, although commonly used as a mood stabilizer, is reported to contribute towards aggressive and agitated behaviors [80]. A systematic review and meta-analysis of the behavioral effects of levetiracetam [81] found the most frequent adverse behavioral effects to be hostility (7.3%), nervousness (6.1%), and aggression (4.9%). The specific choice of anti-seizure medication must take these behavioral side-effects into account, particularly for patients with ASD.

Conversely, there are numerous ways in which anti-seizure medications can improve behavioral symptoms in patients with ASD. Certain anti-seizure medications have mood-stabilizing properties. Selecting those that improve mood can result in lower dosing requirements for anti-depressants or anxiolytics. In patients with intellectual disability, lamotrigine is associated with lower requirements for anti-depressants [82]. Similarly, carbamazepine, valproic acid, and lamotrigine reduce the need for anxiolytics and psychotropics [82]. If aggressive behaviors or SIBs are instead felt to relate to neuropathic pain, consideration may be given to the use of pregabalin or gabapentin [83]. Since SIBs often involve head-hitting [84], headaches should be considered as a potential cause for some SIBs. In patients who are non-verbal, an empiric trial of an anti-headache treatment may be considered. In situations where patients experience both epilepsy and headaches, valproate and topiramate may be used [85]. Topiramate may also be considered for patients with hyperphagia or obesity, since weight loss is a common side-effect [86]. Obesity is more common in ASD patients than in the general population [87] and is a particular concern for ASD patients with aggression who are using anti-psychotic medications [88].

Likewise, treatment choices for aggression need to consider the effects of psychotropic medications on epilepsy. In children aged 2 to 17 years with ASD, only risperidone and aripiprazole are FDA-approved for the treatment of aggression, but they also contribute to side-effects including sedation, extrapyramidal symptoms, and weight gain. Neither of the FDA-approved treatments for ASD and aggression are felt to be helpful in treating epilepsy. They also do not act on underlying inflammatory mechanisms, headaches, or neuropathic pain that may directly contribute to SIB. For ASD patients with epilepsy and aggression, the risk of seizure exacerbation from anti-psychotic use must be taken into account. Although rare, risperidone can exacerbate seizures in patients with epilepsy [89]. Among anti-psychotics, risperidone also has a relatively high probability of inducing EEG abnormalities [90]. Similarly, aripiprazole may exacerbate seizures [91,92] and contribute to epileptiform EEG abnormalities [93]. Clinicians must consider these risks with treating co-morbid epilepsy and aggression.

Such medication interactions speak to the importance of developing treatment trials that examine ASD patients with epilepsy and aggression as a distinct subpopulation (as opposed to ASD, epilepsy, or aggression individually). The limited studies on epilepsy and ASD suggest that this population is distinct from that of patients with epilepsy alone. For example, when examining patients with ASD and epilepsy, generalized motor seizures are more common, and MRI abnormalities are less common than in patients with epilepsy alone [94]. Since the clinical features, laboratory investigations, and treatment responses may differ in patients with ASD, epilepsy, and aggression as compared to those without all three co-morbidities, future research needs to focus on the unique clinical aspects of this subpopulation. Further research is also needed to evaluate the clinical differences and treatment responses in patients with ASD-R versus ASD-NR.

This study is hopefully a first step towards that process. The limitations of this study include its retrospective nature and the small sample size. A larger prospective study is needed to confirm these relationships. Should they be supported, consideration may be given to the pro-active assessment of children with ASD-R for epilepsy and, perhaps, the initiation of anti-seizure medications prior to seizure onset. This may also allow for earlier identification and improved treatment approaches for aggressive behaviors. Should shared etiological underpinnings exist, novel treatment avenues could hopefully be developed which improve all conditions simultaneously. Through understanding ontogeny, treatments can be administered with optimal timing, perhaps even resulting in preventative strategies.

## Figures and Tables

**Table 1 jpm-13-01049-t001:** Ages at autism diagnosis, epilepsy, and aggressive behavior onset, and the prevalence of comorbidities (epilepsy, aggressive behavior, sleep disturbance) in children with ASD-R versus ASD-NR. *p*-Values were calculated using the chi-square test for categorial variables and one-tailed *t*-test for numerical variables that are expressed as the mean (IQR 25–75%). Significant *p*-values (<0.05) are in bold.

Variable		Autism with No Regression, n (%)	Autism with Regression, n (%)	*p*-Values
Total cohort		*n* = 63	*n* = 56	
Sex	F	9 (14.3)	8 (14.3)	1.000
	M	54 (85.7)	48 (85.7)	
Epilepsy	Yes	24 (38.1)	29 (51.8)	0.1337
	No	39 (61.9)	27 (48.2)	
Epilepsy onset age (years)		*n* = 23 (1 missing)	*n* = 29	
Autism Dx age (years)		5.26 (IQR 1.20–10.00)	7.83 (IQR 3.00–12.00)	0.0991
		*n* = 60 (3 missing)	*n*= 54 (2 missing)	
		3.98 (2.00–5.00)	2.46 (1.92–2.83)	0.0011
Aggression and/or SIB	Yes	36 (57.1)	41 (73.2)	0.0673
	No	27 (42.9)	15 (26.8)	
Onset of aggression (years)		*n*= 35 (1 missing)	*n*= 41	
		10.07 (7.50–14.00)	9.40 (7.00–12.00)	0.2211
Sleep disturbance	Yes	33 (52.4)	34 (60.7)	0.3603
	No	30 (47.6)	22 (39.3)	
Comorbid conditions	Epilepsy and aggression	15 (23.8)	25 (44.6)	0.0163
	All others	48 (76.2)	31 (55.4)	

## Data Availability

All data are presented within the article.

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
