# Peer review of "Developmental Regression Followed by Epilepsy and Aggression: A New Syndrome in Autism Spectrum Disorder?"

_jpm, 2023, doi:10.3390/jpm13071049_

Round 1
Reviewer 1 Report
The present study found that aggressive behaviors and epilepsy are more common in patients with ASD who experience regression (ASD-R) than those without regression (ASD-NR), and these comorbidities can significantly impact the quality of life of individuals with ASD and their families. It's contributed significantly to our understand toward ASD. Yet, here are some comments for further improvement of the article,
Introduction
While the literature review provides a comprehensive overview of previous research on comorbidities in ASD, there are some potential areas for improvement.
- The literature review is that it primarily focuses on the prevalence of epilepsy and aggressive behaviors in patients with ASD, but does not provide a detailed discussion of the mechanisms or underlying causes of these comorbidities. While the review briefly touches on the relationship between epilepsy and language regression, it would benefit from a more in-depth exploration of the potential biological, genetic, or environmental factors that may contribute to the development of these comorbidities in ASD.
- Additionally, the literature review could have further explored the potential impact of these comorbidities on the quality of life of individuals with ASD and their families. For example, the review mentions that aggressive behaviors can lead to social isolation and heightened parental stress, but does not provide a detailed discussion of the potential long-term consequences of these challenges.
- Finally, while the literature review provides a clear and concise hypothesis for the study, it could have been strengthened by a more detailed discussion of the potential implications of the study's findings and how they may inform future research or clinical practice.
Materials and Methods
- The study defines ASD-R based on a loss of previously acquired communication or social skills, but it does not provide a detailed description of how this was assessed or measured. This lack of specificity may limit the reliability and validity of the ASD-R diagnosis.
- While the study provides clear definitions for epilepsy and aggression, it is not clear how sleep disruption was defined or assessed. This lack of specificity may limit the reliability and validity of the sleep disruption data.
- Finally, while the statistical analyses are appropriate for the study design and research questions, the study only reports p-values for statistical significance. It would have been helpful to also report effect sizes or confidence intervals to provide a better understanding of the magnitude and precision of the observed differences between groups.
Results
- While the study reports statistically significant differences in the prevalence of epilepsy and aggressive behaviors between ASD-R and ASD-NR patients, it does not provide a detailed discussion of the clinical significance of these findings. It would be helpful to provide more context around the magnitude of the observed differences and how they may impact clinical decision-making or patient outcomes.
- Finally, while the study reports the prevalence of comorbidities, it does not provide a detailed analysis of potential interactions or relationships between these comorbidities. For example, the study reports a higher prevalence of comorbid epilepsy and aggression in ASD-R patients, but it does not explore potential mechanisms or underlying factors that may contribute to this relationship.
Discussion
The discussion section of the study provides a clear summary of the study's findings and their potential implications. However, there are some areas for improvement in the discussion.
- One potential limitation of the discussion is that it does not provide a detailed discussion of the potential mechanisms or underlying factors that may contribute to the observed relationship between ASD-R and comorbid epilepsy and aggression. A more in-depth exploration of potential biological, genetic, or environmental factors could help to shed light on the underlying causes of these comorbidities.
- While the discussion acknowledges the potential impact of epilepsy and aggression on the quality of life of individuals with ASD and their families, it does not provide a detailed discussion of potential interventions or treatments that may be effective in managing these comorbidities. A more in-depth exploration of potential treatment approaches could help to inform clinical decision-making and improve patient outcomes.
- Finally, while the discussion highlights the potential implications of the study's findings for future research and clinical practice, it does not provide a detailed exploration of potential avenues for future research. A more in-depth discussion of potential research questions and approaches could help to guide future studies in this area.
N/A
Author Response
To the editors of Personalized Medicine,
I appreciate the thoughtful input from the reviewers and have incorporated changes to the manuscript based on their feedback. With their input, this is a much stronger work than the initial iteration. Below is a response to each point raised by the reviewers.
Sincerely,
John Gaitanis
“While the review briefly touches on the relationship between epilepsy and language regression, it would benefit from a more in-depth exploration of the potential biological, genetic, or environmental factors that may contribute to the development of these comorbidities in ASD.”
A detailed discussion of possible etiologic mechanisms was added to the discussion section.
- Additionally, the literature review could have further explored the potential impact of these comorbidities on the quality of life of individuals with ASD and their families. For example, the review mentions that aggressive behaviors can lead to social isolation and heightened parental stress, but does not provide a detailed discussion of the potential long-term consequences of these challenges.
Specific examples of aggressive behaviors were added to the methods section. There is also a more complete description of the impact to families added to the introduction.
- Finally, while the literature review provides a clear and concise hypothesis for the study, it could have been strengthened by a more detailed discussion of the potential implications of the study's findings and how they may inform future research or clinical practice.
Further description on this was added to the discussion section—it reviews the importance of developing treatment trials for populations with ASD, epilepsy, and aggression and the importance of using medications that simultaneously address all co-morbidities together.
Materials and Methods
- The study defines ASD-R based on a loss of previously acquired communication or social skills, but it does not provide a detailed description of how this was assessed or measured. This lack of specificity may limit the reliability and validity of the ASD-R diagnosis.
Detailed descriptions for regression were added to the methods sections, which included specific patient examples.
- While the study provides clear definitions for epilepsy and aggression, it is not clear how sleep disruption was defined or assessed. This lack of specificity may limit the reliability and validity of the sleep disruption data.
A more detailed description of sleep disturbance was added to the methods section
- Finally, while the statistical analyses are appropriate for the study design and research questions, the study only reports p-values for statistical significance. It would have been helpful to also report effect sizes or confidence intervals to provide a better understanding of the magnitude and precision of the observed differences between groups.
Confidence intervals were added to the results section.
Results
- While the study reports statistically significant differences in the prevalence of epilepsy and aggressive behaviors between ASD-R and ASD-NR patients, it does not provide a detailed discussion of the clinical significance of these findings. It would be helpful to provide more context around the magnitude of the observed differences and how they may impact clinical decision-making or patient outcomes.
A much more detailed description of the impact to decision making was added to the discussion section.
- Finally, while the study reports the prevalence of comorbidities, it does not provide a detailed analysis of potential interactions or relationships between these comorbidities. For example, the study reports a higher prevalence of comorbid epilepsy and aggression in ASD-R patients, but it does not explore potential mechanisms or underlying factors that may contribute to this relationship.\
A much more detailed description of the mechanisms and underlying causes were comorbid epilepsy and aggression was added to the discussion section.
Discussion
- One potential limitation of the discussion is that it does not provide a detailed discussion of the potential mechanisms or underlying factors that may contribute to the observed relationship between ASD-R and comorbid epilepsy and aggression. A more in-depth exploration of potential biological, genetic, or environmental factors could help to shed light on the underlying causes of these comorbidities.
An in-depth discussion on genetic, metabolic, an immunologic contributions to ASD-R, epilepsy, and aggression was added to the discussion.
- While the discussion acknowledges the potential impact of epilepsy and aggression on the quality of life of individuals with ASD and their families, it does not provide a detailed discussion of potential interventions or treatments that may be effective in managing these comorbidities. A more in-depth exploration of potential treatment approaches could help to inform clinical decision-making and improve patient outcomes.
A more detailed discussion on the interaction of epilepsy and psychiatric medications was added to the discussion section.
- Finally, while the discussion highlights the potential implications of the study's findings for future research and clinical practice, it does not provide a detailed exploration of potential avenues for future research. A more in-depth discussion of potential research questions and approaches could help to guide future studies in this area.
The need for increased study on this topic was included to the end of the discussion section.
Reviewer 2 Report
This is retrospectve study in which Authors analyzed the risk of having epilepsy and aggression in the to context of ASD linked to regression. It is interesting study. The introduction is detailed, described comprehensively background. Authors gave resoanle conclusions and they also mentioned limitations of the study. I have only one comment: in table 1 there two abbr. SIB and Dx. What they mean? Once explained, paper is ready for publication.
Author Response
To the editors of Journal of Personalized Medicine,
I appreciate the thoughtful input from the reviewers. I believe their comments have made this a much stronger paper. Below is my response to the reviewer’s suggestions.
Best,
John Gaitanis
Reviewer #2:
This is retrospectve study in which Authors analyzed the risk of having epilepsy and aggression in the to context of ASD linked to regression. It is interesting study. The introduction is detailed, described comprehensively background. Authors gave resoanle conclusions and they also mentioned limitations of the study. I have only one comment: in table 1 there two abbr. SIB and Dx. What they mean? Once explained, paper is ready for publication.
The abbreviations in the table were removed and replaced with the full text of “self-injurious behavior” and “diagnosis. “